# From physical self-esteem to sports participation: The mediating role of exercise motivation and social support in adolescents

Bo Peng[1], Weisong Chen[1], Hongshen Wang🔘[1]*, Ting Yu[2]

**1** Sports Training Academy, Chengdu Sport University, Chengdu, Sichuan, China, **2** Jingshan Primary School, Changshou, Chongqing, China

\* hongshen1101@163.com

## Abstract

### Objective

This study examines the mechanisms linking physical self-esteem to adolescents' sports participation, emphasizing the mediating roles of exercise motivation and social support. It aims to establish a comprehensive framework that integrates psychological and social factors to understand sports behavior among adolescents.

### Method

A total of 2,588 adolescents from various regions in China were surveyed using validated scales for physical self-esteem, exercise motivation, social support, and sports participation. Structural equation modeling (SEM) was used to assess direct, mediated, and sequential mediation effects.

### Results

The analysis confirmed that physical self-esteem significantly predicts sports participation directly ($\beta = 0.094$, $p < 0.001$). Exercise motivation and social support act as mediators, explaining 33.26% and 16.74% of the total effect, respectively. Sequential mediation analysis showed that physical self-esteem influences sports participation through the combined effects of exercise motivation and social support, contributing 29.61% to the total effect. Demographic analyses revealed that males, rural adolescents, and middle school students exhibit higher levels of self-esteem, motivation, and sports participation. Group differences were analyzed based on demographic variables, including gender, grade level, and family location.

### Conclusion

This study underscores the critical roles of psychological traits and social contexts in shaping adolescents' sports participation. By revealing the mediating and sequential

**Data availability statement:** The original contributions presented in the study are included in the article/supplementary material.

**Funding:** The author(s) received no specific funding for this work.

**Competing interests:** The authors have declared that no competing interests exist.

mediation effects of exercise motivation and social support, the research highlights pathways for targeted interventions. Schools and communities should focus on enhancing physical self-esteem and fostering supportive social networks to encourage sports participation. Future studies should incorporate longitudinal designs and explore cultural and regional variations to extend these findings.

## 1. Introduction

Adolescence is a stage of rapid physical and psychological development, and physical self-esteem is an important component of an individual's self-perception, which has a profound impact on behavior and mental health [1–3]. Physical self-esteem not only reflects adolescents' evaluation of their own body image but is also closely related to their emotions, social interactions, and quality of life [4,5]. Studies have shown that physical self-esteem plays a key role in adolescents' social adaptation, emotional regulation, and health behaviors, especially in terms of sports participation [6–8]. Sports participation is considered an important way to promote adolescents' physical and mental health, as it helps to improve physical fitness, enhance mood, and boost social abilities [9,10]. However, despite the widespread exploration of the relationship between physical self-esteem and sports participation, the specific mechanisms remain insufficiently elucidated.

In sports activities, exercise motivation and social support are two important psychological and social factors, both playing a significant role in adolescents' sports participation [11,12]. Exercise motivation determines an individual's intrinsic drive and persistence in engaging in sports activities [13], and research has shown that adolescents with higher intrinsic motivation are more likely to continue participating in sports and experience greater psychological and physical benefits [14,15]. Meanwhile, social support—especially emotional support from family, peers, and coaches—is considered to provide essential encouragement and motivation, enhancing adolescents' confidence and willingness to engage in sports activities [16–18]. Although existing research has gradually revealed the role of exercise motivation and social support in sports participation [19,20], research on how these three factors (physical self-esteem, exercise motivation, and social support) jointly influence sports participation remains insufficient.

Current literature mainly focuses on the direct effects of physical self-esteem on sports participation [21], or separately examines the roles of exercise motivation and social support [19,20]. However, the systematic exploration of how these three factors collaborate in adolescent sports behavior has not been fully addressed. Particularly in the context of an increasingly diverse adolescent population, understanding how physical self-esteem, exercise motivation, and social support interact and influence sports participation remains an area worth further investigation.

Therefore, this study aims to explore how adolescents' physical self-esteem influences their sports participation through the dual pathways of exercise motivation and social support. Unlike traditional studies, this research attempts to integrate these

three factors into a comprehensive model, revealing their complex relationships in adolescents' sports participation. This study not only helps to fill the gap in current literature but also provides new perspectives and theoretical support for strategies promoting adolescents' sports participation.

## 2. Literature review and research hypotheses

### 2.1. The relationship between adolescent physical self-esteem and sports participation

Physical self-esteem is a complex and multidimensional concept that encompasses various components, including perception, cognition, emotion, and behavior [22]. Its perceptual dimension is typically referred to as body self-esteem, which is defined as an individual's subjective evaluation of their body [23]. Originally, body self-esteem was defined as the evaluation of various aspects of one's body within the framework of social judgment; as a part of overall self-esteem, it includes both a general sense of self-worth regarding the body and more specific satisfaction with various bodily aspects, involving individual cognitive, emotional, and evaluative processes [24]. Some researchers have further defined body self-esteem as "the satisfaction or dissatisfaction with various aspects of one's physical appearance," emphasizing its close relationship with social evaluation, and considering it an important component of self-esteem [25]. Additionally, other scholars have viewed body self-esteem more positively, associating it with a positive body image, which involves the emotions evoked by an individual's subjective perception and evaluation of their body, and emphasizing the importance of appreciating and caring for one's body [26].

During adolescence, body self-esteem, as an important component of self-concept, influences psychological state and behavioral performance [27]. Higher body self-esteem is often associated with more positive behavioral choices in adolescents, particularly in terms of participation in physical activities[28,29]. Specifically, body self-esteem enhances adolescents' initiative and persistence in sports by boosting their self-confidence and self-acceptance[30,31]. Additionally, body self-esteem helps reduce anxiety and self-rejection caused by dissatisfaction with body image, thereby increasing adolescents' interest in and psychological readiness for sports participation [32–34]. Furthermore, body self-esteem plays a crucial role in emotional regulation and the enhancement of self-efficacy, with these factors collectively influencing adolescents' attitudes and behaviors toward physical activities [35–37]. Through these psychological mechanisms, body self-esteem provides positive psychological resources that can impact adolescents' performance and sustained engagement in sports [38].

### 2.2. Mediating variables in the relationship between adolescent physical self-esteem and sports participation

**2.2.1. Exercise motivation.** Motivation is the intrinsic drive that prompts individuals to engage in a particular activity. It stimulates the willingness to participate and reinforces the individual's behavioral motivation on a psychological level [39]. In psychology, the initiation and maintenance of behavior are driven by continuous motivation. Motivation not only reflects the effort an individual exerts to achieve goals but also represents the primary drive formed to meet spiritual, physical, or social needs [40]. The formation of motivated behavior is closely linked to autonomy needs, with motivation existing at three levels: amotivation, extrinsic motivation, and intrinsic motivation. Extrinsic motivation includes internalization, identification, and external regulation, while intrinsic motivation encompasses curiosity, achievement, and stimulation. Research shows that the stronger the intrinsic motivation, the greater the individual's interest, motor skills, and learning outcomes in sports. Therefore, the different levels of motivation have significant effects on sports outcomes and psychological states [40]. In this study, exercise motivation is defined as the intrinsic drive and external factors that prompt individuals to participate in sports activities, encompassing their interest, desire, and goals related to sports. Exercise motivation encourages individuals to actively engage in physical exercise and participate in various sports events.

Physical self-esteem indirectly influences adolescents' exercise motivation by enhancing their confidence and self-acceptance [30,31]. Adolescents with higher levels of self-esteem tend to view sports activities as a way to enhance their self-identity and self-efficacy, which provides psychological support and stimulates stronger exercise motivation [41,42].

In addition, physical self-esteem helps regulate emotions and reduces negative feelings arising from dissatisfaction with body image, improving emotional regulation abilities and creating favorable conditions for enhancing exercise motivation [37,43]. Physical self-esteem is also closely related to self-efficacy, and higher levels of self-esteem boost adolescents' confidence in sports activities, further promoting their exercise motivation and sustained participation [44–46]. The enhancement of exercise motivation plays an important role in sports participation. Studies have shown that stronger intrinsic motivation significantly increases the frequency and quality of sports participation [47,48]. Therefore, physical self-esteem, by boosting exercise motivation, provides essential psychological support for adolescents' sports participation, which may, in turn, impact the sustainability and effectiveness of their long-term engagement.

**2.2.2. Social support.** Social support, especially from family, peers, and coaches, provides emotional security and encouragement for adolescents, thereby influencing their sports participation [16,17]. Social support enhances an individual's sense of belonging and self-worth, which helps to increase adolescents' interest and engagement in sports activities, further stimulating their intrinsic motivation [49–51]. Specifically, supportive interactions from family and peers can emotionally alleviate adolescents' anxiety and unease, enabling them to demonstrate greater confidence and willingness to participate in sports [52,53]. Additionally, social support improves emotional regulation and self-efficacy, strengthening an individual's ability to persist and cope with challenges, thus promoting exercise motivation and long-term participation [54]. Research has shown that supportive social networks not only improve adolescents' emotional well-being but also promote sustained sports participation by optimizing their social environment [55,56]. Therefore, social support may play a crucial role in adolescents' sports participation by regulating exercise motivation, although the specific mechanisms of this process still require further exploration.

## 2.3. Chain mediation effect

Physical self-esteem significantly influences sports participation by affecting adolescents' exercise motivation and social support. High levels of physical self-esteem enhance adolescents' confidence and self-acceptance [57,58], which makes them more likely to view sports activities as a way to improve self-efficacy and self-identity, thereby increasing their motivation to participate in sports [30,31]. Moreover, improvements in physical self-esteem lead to greater emotional stability and increase adolescents' opportunities for positive feedback in sports [59–61]. As adolescents receive more emotional support, particularly from family and peers, their exercise motivation is further enhanced, creating a positive cycle of sports participation [62].

At the same time, social support plays a crucial role in this process. Adolescents with high self-esteem are more likely to effectively utilize encouragement and support from others, which not only increases their willingness to engage in sports but also improves their attitude toward sports participation [63,64]. Emotional support and social recognition from others provide adolescents with additional motivation, helping them overcome difficulties and challenges in sports, and promoting sustained involvement in physical activities [65]. Therefore, understanding the interaction between physical self-esteem, exercise motivation, and social support may offer new perspectives for promoting adolescents' sports participation, although this mechanism still needs further validation in empirical research.

## 2.4. Theoretical model

Based on the existing literature on the relationships between physical self-esteem, exercise motivation, social support, and adolescent sports participation, this study integrates relevant theoretical and empirical research findings to propose a comprehensive theoretical framework (see Fig 1), aimed at exploring the following aspects:

1. **Direct relationship:** The direct impact of adolescent physical self-esteem on sports participation.

2. **Mediating effect:** The mediating role of exercise motivation and social support in the relationship between physical self-esteem and sports participation.

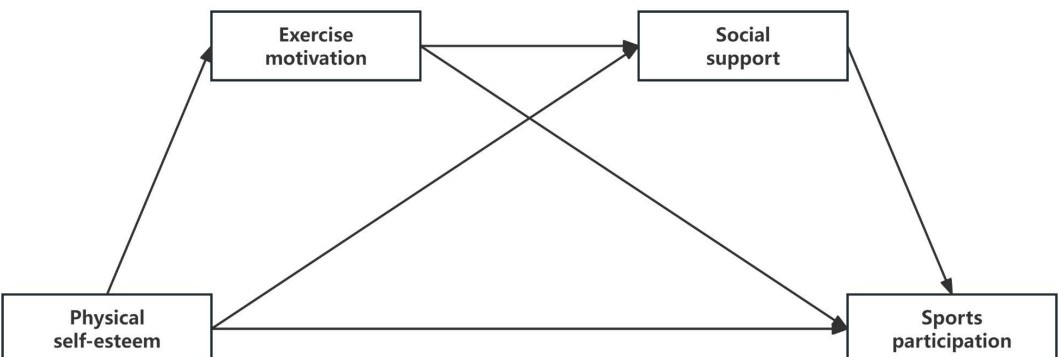

**Fig 1. Theoretical model of the impact of adolescent physical self-esteem on sports participation: the chain mediation effect of exercise motivation and social support.**

3. **Chain mediation effect:** The interaction between exercise motivation and social support, highlighting how these two factors jointly influence the sustainability and extent of adolescents' sports participation.

This model hypothesizes that adolescent physical self-esteem influences sports participation directly or indirectly through factors such as exercise motivation and social support. Exercise motivation and social support may play important mediating roles in this process, facilitating the impact of physical self-esteem on sports participation. By integrating these factors, this framework provides a systematic theoretical perspective on the multidimensional influences on adolescents' sports participation, and offers theoretical support and practical guidance for developing targeted interventions.

Based on the above, the hypotheses of this study are as follows:

**Hypothesis 1 (H1):** Physical self-esteem significantly positively predicts adolescents' sports participation.

**Hypothesis 2 (H2):** Exercise motivation mediates the relationship between physical self-esteem and sports participation in adolescents.

**Hypothesis 3 (H3):** Social support mediates the relationship between physical self-esteem and sports participation in adolescents.

**Hypothesis 4 (H4):** Exercise motivation and social support jointly mediate the relationship between physical self-esteem and sports participation in adolescents.

## 3. Materials and methods

### 3.1. Participants and data

**3.1.1. Sample size justification.** To ensure the robustness of the research design and the reliability of the results, this study estimated the required sample size using semPower for structural equation modeling (SEM) and the rule of thumb commonly applied in social science research.

First, a power analysis for SEM was conducted using semPower[66], with the following parameters: desired power of 0.80, alpha level of 0.05, RMSEA as the effect measure, an effect size of 0.05, degrees of freedom (df) of 50, and 12 manifest variables. Under these conditions, the analysis indicated that at least 243 participants would be needed to detect a significant effect in the structural equation model. semPower analysis ensures that, with the preset statistical power, the relationships among physical self-esteem, exercise motivation, social support, and sports participation can be accurately assessed.

   

Second, according to the rule of thumb in social science research, it is recommended that the sample size be 10–15 times the total number of items in the questionnaire. The measurement tools used in this study included: the physical self-esteem scale (30 items), the sports participation scale (3 items), the exercise motivation scale (28 items), and the social support scale (12 items). Therefore, the estimated sample size range was between 730 and 1,095 participants (i.e., 73 items × 10–73 items × 15). This method further ensures that the sample size is scientifically appropriate and comprehensive in the actual survey design.

Based on these two methods, the minimum required sample size for this study was 243 participants based on SEM power analysis, while the ideal sample range based on the questionnaire rule of thumb was 730–1,095 participants. During the actual data collection process, to enhance the statistical power and external validity of the research results, data from 2,588 adolescent participants were ultimately collected. This large sample size significantly exceeds the minimum requirements set by both the power analysis and the rule of thumb, providing sufficient data support for analyzing the relationships among the research variables. Furthermore, exceeding the minimum sample size requirement improves the stability, generalizability, and representativeness of the findings across different demographic characteristics.

**3.1.2. Participant selection process.** The participants in this study were students from grades 7–12, including both junior high school (grades 7–9) and senior high school (grades 10–12) students. Elementary school students (grades 1–6) were excluded due to their limited ability to comprehend the questionnaire. In order to enhance representativeness and minimize sampling bias, a multi-stage cluster sampling method was employed. Stratification was performed based on school level (junior vs. senior high school), gender, and family location (urban vs. rural), ensuring balanced representation across these groups.

The sample was drawn from schools across various regions of China. Initially, several schools were randomly selected from each region. Then, within each selected school, one or two classes per grade were randomly chosen, and all students in those classes were invited to participate in the study. The inclusion criteria for participation were: students enrolled in grades 7–12, the ability to independently complete the questionnaire, and regular engagement in physical exercise. During the data cleaning process, invalid or incomplete questionnaires (e.g., those with missing data or showing response patterns indicative of non-serious responses) were excluded from the final dataset.

To evaluate potential clustering effects due to the multi-stage sampling design, intra-class correlation coefficients (ICCs) were calculated for all key study variables. The results revealed minimal clustering influence: School-level ICC values ranged from 0.5% to 1.1%, with physical self-esteem at 0.8% and sports participation at 0.5%. Classroom-level ICC values ranged from 0.9% to 1.8%, with social support at 1.8% and exercise motivation at 1.2%.

To account for any residual clustering effects in our analyses, we employed Huber-White robust standard errors. This approach ensures that the relationships among the key variables—physical self-esteem, exercise motivation, social support, and sports participation—are accurately estimated, accounting for potential correlations within schools and classrooms.

**3.1.3. Data collection method.** To ensure the validity and reliability of the data, this study employed strict standardized procedures for data collection. The specific process is as follows:

*Training of data collectors:* Before distributing the questionnaires, the research team conducted systematic training for the physical education teachers and homeroom teachers who were involved in data collection. The training covered the explanation of the research objectives, the importance of random sampling, and the specific procedures for standardized questionnaire distribution. Through detailed guidance and situational simulations, the training ensured that data collectors could implement the questionnaire distribution and monitoring processes consistently, thereby minimizing the potential impact of human error on data quality.

*Questionnaire distribution and completion*: The distribution and completion of the questionnaires were carried out in a controlled environment during regular school hours, ensuring a quiet, undisturbed setting. Physical education teachers and homeroom teachers supervised the entire process to ensure that students could independently complete the

questionnaires without external influence. Once the questionnaires were completed, the data were immediately collected and securely sealed to avoid delays or possible content leaks, ensuring the timeliness and accuracy of the data.

***Confidentiality and ethical approval***: This study strictly adhered to the Declaration of Helsinki and the ethical guidelines of relevant national and institutional authorities. The research plan was reviewed and approved by the Ethics Committee of Chengdu Sport University (Approval Number: CTYLL2024017). Since all participants were minors, dual consent was obtained prior to participation: verbal informed consent from the students and written consent from their parents or guardians. In certain cases, due to logistical or access constraints, where written parental consent could not be directly obtained, the Ethics Committee approved a waiver based on specific circumstances, ensuring the study was conducted within an ethically compliant framework.

***Data privacy and anonymity***: All data were processed anonymously to ensure participants' privacy was fully protected. The questionnaires did not collect any personally identifiable information, and strict confidentiality measures were adhered to throughout the data analysis process to safeguard participants' rights. Additionally, the research team implemented encryption protection measures during the data storage and sharing stages to further ensure data security.

**3.1.4. Data processing.** The survey lasted for 5 months, starting on April 1, 2024, and concluding on September 1, 2024. A total of 2,900 questionnaires were distributed. To ensure data accuracy and research rigor, the returned questionnaires underwent meticulous screening. Invalid questionnaires, such as those with missing responses, incorrect answers, or fixed-pattern selections, were excluded. Ultimately, 2,588 valid questionnaires were retained, resulting in a high effective response rate of 89.24%. During the screening process, strict predefined criteria were applied to exclude incomplete responses or questionnaires with inconsistent answers. This ensured the high quality and completeness of the data. Detailed information about the respondents is provided in Table 1.

## 3.2. Measurement

All the measurement tools used in this study are based on existing scales that have been validated and demonstrated good reliability and validity. The specific measurement tools and related information are as follows:

***Physical self-esteem***: Physical self-esteem was assessed using a scale developed by Xu Xia et al. [67]. This scale consists of 5 dimensions and 30 items, aimed at evaluating individuals' self-evaluation and attitudes towards their physical appearance, body function, and body acceptance. The scale uses a 4-point Likert scale, ranging from 1 ("completely disagree") to 4 ("completely agree"), with higher scores indicating stronger physical self-esteem. Previous studies have shown that the scale exhibits good reliability (Cronbach's $\alpha > 0.80$) and validity across different cultures and populations. In this study, the Cronbach's $\alpha$ for this scale was 0.952.

***Sports participation***: Sports participation was assessed using a scale developed by Liang Deqing et al. [68]. This scale consists of 3 items, aimed at measuring the frequency and intensity of adolescents' participation in sports activities. The scale uses a 5-point Likert scale, ranging from 1 ("very poor") to 5 ("very good"), with higher scores indicating higher

**Table 1. The sample information.**

| Basic information | Category | Frequency | Percentage |
|---|---|---|---|
| Gender | Male | 1569 | 60.63% |
| | Female | 1019 | 39.37% |
| School stage | Middle school | 1318 | 50.93% |
| | High school | 1270 | 49.07% |
| Residence type | Rural | 922 | 35.63% |
| | Urban | 1666 | 64.37% |

levels of sports participation. Previous studies have demonstrated that the scale has good reliability (Cronbach's α > 0.80) and validity across different cultures and populations. In this study, the Cronbach's α for this scale was 0.803.

*Exercise motivation*: Exercise motivation was assessed using a scale developed by Zhang Liwei et al. [69]. This scale consists of 3 dimensions and 28 items, aimed at evaluating individuals' motivation for engaging in physical activity or sports, uncovering the psychological and behavioral factors that drive individuals to participate in physical activities. The scale uses a 7-point Likert scale, ranging from 1 ("completely disagree") to 7 ("completely agree"), with higher scores indicating stronger exercise motivation. Previous studies have shown that the scale has good reliability (Cronbach's α > 0.80) and validity across different cultures and populations. In this study, the Cronbach's α for this scale was 0.962.

*Social support*: Social support was assessed using the social support scale developed by Chen Wei et al. [70]. This scale consists of 3 dimensions and 12 items, aimed at evaluating the extent of support individuals receive in social interactions and their subjective feelings of support. The scale uses a 5-point Likert scale, ranging from 1 ("completely disagree") to 5("completely agree"), with higher scores indicating greater support received. Previous studies have shown that the scale has good reliability (Cronbach's α > 0.80) and validity across different cultures and populations. In this study, the Cronbach's α for this scale was 0.908.

*Summary*: Table 2 summarizes the measurement tools used in this study, including the number of items, scoring range, reliability, and standard deviations. These scales were selected based on their widely validated stability and applicability in relevant fields, effectively capturing the key characteristics of the core variables in this research.

### 3.3. Data analysis procedures

Data analysis was conducted using SPSS 26.0 and AMOS 24.0. Descriptive statistics, including means and standard deviations for all key variables, were calculated to provide an overview of the sample's characteristics and data distribution. These metrics provided foundational insights for further statistical testing and ensured the data met the basic requirements for subsequent analyses. Cronbach's α coefficients were used to assess the reliability of each construct, with an acceptable reliability threshold of 0.70 and a strong internal consistency threshold of 0.80 [71].

Correlation analysis was performed to examine the relationships between key variables. Pearson correlation coefficients were used to determine the strength and direction of these associations, with the coefficients interpreted as follows: 0.10–0.29 (weak), 0.30–0.49 (moderate), ≥ 0.50 (strong). This analysis provided preliminary evidence for the structural relationships to be tested in subsequent modeling [72].

**Table 2. Scales used in this study.**

| Scale | Author (Year) | Item quantity | Scoring | Dimensions |
|---|---|---|---|---|
| Physical self-esteem | Xia Xu(2001) | 30 | 4 | Physical self-worth; Sport ability; Physical condition; Physical attractiveness; Physical fitness |
| Physical exercise | Deqing Liang (1994) | 3 | 5 | Exercise intensit; Duratio; exercise Frequency |
| Exercise motivation | Liwei Zhang (2004) | 28 | 7 | Intrinsic motivation; Extrinsic motivation; Amotivation |
| Social support | Wei Chen(2016) | 12 | 5 | Family; Friends; Others |

To explore potential differences between subgroups, independent sample t-tests and one-way analysis of variance (ANOVA) were conducted to assess differences in physical self-esteem, exercise motivation, social support, and sports participation based on demographic variables such as gender, grade level, and family location [73].

To assess potential common method bias, Harman's single-factor test was conducted using principal component analysis (PCA) [72]. All survey items were loaded into an unrotated factor solution, and the proportion of variance explained by the largest factor was evaluated against the critical threshold of 40%. This ensured that common method bias was minimized and did not interfere with subsequent analyses.

Structural Equation Modeling (SEM) was used to assess the hypothesized direct and mediation effects. Model fit was evaluated using the following indices and thresholds: $\chi^2$/df: < 3.00 (acceptable fit), CFI: ≥ 0.95 (excellent fit), TLI: ≥ 0.95 (excellent fit), RMSEA: ≤ 0.06 (excellent fit), SRMR: ≤ 0.08 (acceptable fit). Path analysis within the SEM framework examined both direct effects (e.g., physical self-esteem→sports participation) and mediation effects through constructs such as exercise motivation and social support. Standardized coefficients (β) were reported to quantify the strength of these relationships, with significance determined at $p < 0.05$ [74].

To confirm the mediation effects, bootstrapping with 2,000 resamples was performed. Bias-corrected confidence intervals (95%) were generated, and mediation effects were considered significant if the confidence intervals did not include zero. Standardized coefficients were used to calculate effect sizes for direct, indirect, and total effects, providing a comprehensive understanding of the pathways in the model [75].

## 4. Result

### 4.1. Common method bias test

To rigorously assess potential common method bias (CMB), we employed the Unmeasured Latent Method Construct (ULMC) approach within a structural equation modeling (SEM) framework, following the recommendations of Podsakoff et al. (2012) [76]. This method improves upon traditional single-factor tests by explicitly modeling method variance through an additional latent factor that affects all observed indicators. We first established a confirmatory factor analysis (CFA) model aligning with the theoretical constructs of the study, then added a method factor to the model with paths constrained to equality across all indicators to avoid overfitting. Covariances between the method factor and the substantive factors were fixed at zero to ensure model parsimony and interpretability.

The comparison between the Baseline Model and the ULMC Model revealed negligible improvement, suggesting that common method bias does not substantially distort the findings. Specifically, the ΔCFI was 0.007 (Baseline CFI = 0.983 vs. ULMC CFI = 0.990), and the ΔRMSEA was 0.002 (Baseline RMSEA = 0.040 vs. ULMC RMSEA = 0.038). Additionally, the method factor accounted for 13.2% of the total variance, indicating minimal method bias. These results collectively support that common method bias does not substantially affect the relationships in our model, providing confidence in the robustness of our findings.

### 4.2. Descriptive statistics, reliability, and construct validity of the measurement model

Table 3 summarizes the descriptive statistics, internal consistency reliability, and fit indices for confirmatory factor analysis (CFA) of the key variables. The mean for physical self-esteem was 2.819 (SD = 0.613), while sports participation had the highest mean (25.516, SD = 22.334), indicating substantial variability in participation. The mean values for exercise motivation and social support were 4.554 (SD = 1.309) and 3.613 (SD = 0.781), respectively.

Cronbach's α values for all constructs exceeded 0.7, indicating good internal consistency, with particularly high reliability for physical self-esteem (α = 0.952) and exercise motivation (α = 0.962). The reliability for sports participation was α = 0.803, which meets the acceptable threshold.

The results of the confirmatory factor analysis (CFA) demonstrated good model fit, with the Comparative Fit Index (CFI) and Tucker-Lewis Index (TLI) falling within the ideal range. The Standardized Root Mean Square Residual (SRMR) was

**Table 3. Descriptive statistics, internal consistency reliability, and fit indices for confirmatory factor analysis (CFA) of key variables.**

| Variable | M | SD | α | CFI | TLI | SRMR | RMSEA (90%CI) |
|---|---|---|---|---|---|---|---|
| Physical self-esteem | 2.819 | 0.613 | 0.952 | 0.977 | 0.975 | 0.020 | 0.034 (0.032-0.035) |
| Sports participation | 25.516 | 22.334 | 0.803 | – | – | – | – |
| Exercise motivation | 4.554 | 1.309 | 0.962 | 0.980 | 0.978 | 0.016 | 0.037 (0.035-0.038) |
| Social support | 3.613 | 0.781 | 0.908 | 0.990 | 0.987 | 0.018 | 0.038 (0.033-0.043) |

below 0.03, and the Root Mean Square Error of Approximation (RMSEA) was also below 0.05, indicating excellent construct validity.

Table 4 compares alternative factor structures to assess model robustness. The four-factor model, comprising physical self-esteem (PSE), exercise motivation (EM), social support (SS), and sports participation (SP), demonstrated the best fit ($\chi^2 = 363.16$, df = 71, CFI = 0.980, TLI = 0.974, SRMR = 0.025, RMSEA = 0.040 [90% CI = 0.036–0.044]). In contrast, the three-factor, two-factor, and one-factor models showed significantly worse fit indices. For example, the one-factor model exhibited the weakest fit ($\chi^2 = 3722.29$, df = 77, CFI = 0.749, TLI = 0.703, SRMR = 0.093, RMSEA = 0.135 [90% CI = 0.132–0.139]). The $\Delta\chi^2$ test indicated that the four-factor model was significantly better than the alternative models at the 0.01 level.

These results confirm the reliability and validity of the measurement model, supporting its suitability for further structural model analysis.

## 4.3. Correlation analysis among key variables

The correlation analysis among key variables was conducted, and the results are presented in Fig 2. The correlation matrix highlights the relationships between physical self-esteem (PSE), exercise motivation (EM), social support (SS), and sports participation (SP).

Significant positive correlations were observed between all variables, with correlation coefficients ranging from 0.39 to 0.59. Specifically: Physical self-esteem (PSE) showed a moderate positive correlation with exercise motivation (EM; $r = 0.45$, $p < 0.001$), social support (SS; $r = 0.39$, $p < 0.001$), and sports participation (SP; $r = 0.43$, $p < 0.001$). Exercise motivation (EM) exhibited a strong positive correlation with social support (SS; $r = 0.51$, $p < 0.001$) and sports participation (SP; $r = 0.56$, $p < 0.001$). Social support (SS) demonstrated a significant positive correlation with sports participation (SP; $r = 0.59$, $p < 0.001$).

These results indicate that higher levels of physical self-esteem are associated with greater exercise motivation, higher social support, and increased sports participation. Furthermore, exercise motivation and social support are strongly interrelated and positively linked to sports participation, suggesting potential mediating roles for these variables in the relationship between physical self-esteem and sports participation.

**Table 4. Comparative fit indices for alternative factor structures of key constructs.**

| Model | Factor | χ2 | df | Delta χ2 (Delta df) | CFI | TLI | SRMR | RMSEA (90%CI) |
|---|---|---|---|---|---|---|---|---|
| Four-factor model | PSE, EM, SS, SP | 363.16 | 71 | – | 0.980 | 0.974 | 0.025 | 0.040 (0.036-0.044) |
| Three-factor model | PSE+EM, SS, SP | 2002.29 | 74 | -0.113 (-0.137) | 0.867 | 0.837 | 0.078 | 0.100 (0.097-0.104) |
| Two-factor model | PSE+EM+SS, SP | 3186.75 | 76 | -0.194 (-0.231) | 0.786 | 0.743 | 0.089 | 0.126 (0.122-0.130) |
| One-factor model | PSE+EM+SS+SP | 3722.29 | 77 | -0.231 (-0.271) | 0.749 | 0.703 | 0.093 | 0.135 (0.132-0.139) |

PSE, Physical Self-Esteem; SP, Sports participation; EM, Exercise Motivation; SS, Social Support. All Δχ2 passed the significance test at 0.01 level.

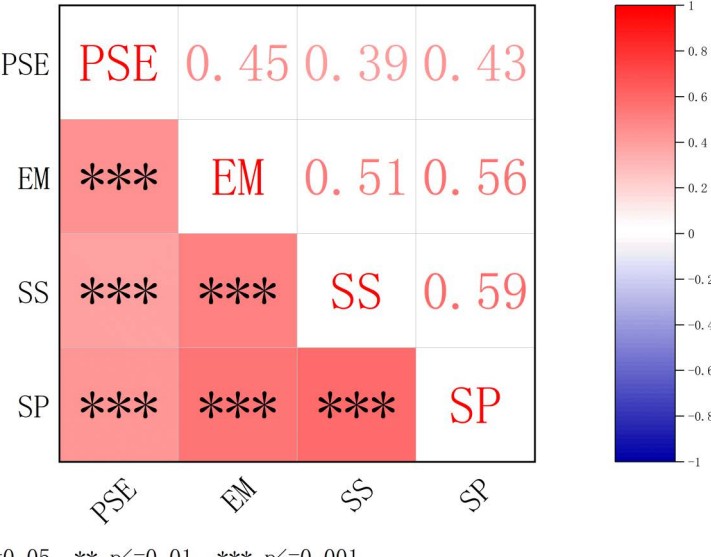

* p<=0.05   ** p<=0.01   *** p<=0.001

**Fig 2. Correlation matrix of key variables.** PSE, Physical Self-Esteem. SP, Sports participation. EM, Exercise Motivation. SS, Social Support.

## 4.4. Differences in key variables among demographic groups of adolescents

Table 5 presents descriptive statistics and t-tests comparing physical self-esteem, exercise motivation, social support, and sports participation across gender, grade level, and family location. These comparisons serve descriptive purposes to provide contextual understanding of the sample's demographic composition, rather than testing substantive hypotheses or implying causal relationships. As such, these analyses are intended to contextualize sample representativeness, not to draw theoretical conclusions [77].

Males reported higher scores than females in all variables, including physical self-esteem (t=7.386, p<0.001), exercise motivation (t=8.066, p<0.001), social support (t=6.873, p<0.001), and sports participation (t=6.914, p<0.001).

High school students scored higher in physical self-esteem (t=−5.575, p<0.001) and exercise motivation (t=−2.936, p=0.003), whereas middle school students reported higher levels of social support (t=9.856, p<0.001) and sports participation (t=9.033, p<0.001).

Adolescents from rural areas had significantly higher scores in all variables compared to urban adolescents, including physical self-esteem (t=−13.018, p<0.001), exercise motivation (t=−16.066, p<0.001), social support (t=−8.415, p<0.001), and sports participation (t=−12.995, p<0.001).

These results emphasize notable demographic differences in key variables, with males, rural adolescents, and middle school students exhibiting higher levels in specific dimensions.

While these results highlight notable demographic differences, it is important to emphasize that these comparisons are descriptive in nature and aim to provide context for understanding the demographic composition of the sample, not to test substantive hypotheses. As such, these analyses are not central to the core research question.

## 4.5. Test results of mediation effects

The mediation effects of exercise motivation and social support on the relationship between physical self-esteem and sports participation in adolescents were analyzed. The results are presented in Table 6, Fig 3, and Table 7.

**Table 5. Descriptive statistics and differences across demographic variables for key variables.**

| Demographic variables | Category | Physical self-esteem | | Exercise motivation | | Social support | | Sports participation | |
|---|---|---|---|---|---|---|---|---|---|
| | | M | SD | M | SD | M | SD | M | SD |
| Gender | Male | 2.890 | 0.612 | 4.720 | 1.274 | 3.699 | 0.746 | 27.892 | 22.952 |
| | Female | 2.709 | 0.597 | 4.300 | 1.322 | 3.481 | 0.815 | 21.858 | 20.835 |
| | t | 7.386 | | 8.066 | | 6.873 | | 6.914 | |
| | P | 0.000 | | 0.000 | | 0.000 | | 0.000 | |
| Grade | Middle school | 2.753 | 0.608 | 4.480 | 1.320 | 3.759 | 0.743 | 29.339 | 23.608 |
| | High school | 2.887 | 0.610 | 4.631 | 1.293 | 3.461 | 0.790 | 21.549 | 20.187 |
| | t | -5.575 | | -2.936 | | 9.856 | | 9.033 | |
| | P | 0.000 | | 0.003 | | 0.000 | | 0.000 | |
| Family location | Urban | 2.614 | 0.595 | 4.017 | 1.289 | 3.442 | 0.780 | 18.421 | 19.296 |
| | Rural | 2.932 | 0.593 | 4.852 | 1.223 | 3.708 | 0.765 | 29.443 | 22.930 |
| | t | -13.018 | | -16.066 | | -8.415 | | -12.995 | |
| | P | 0.000 | | 0.000 | | 0.000 | | 0.000 | |

**Table 6. Questionnaire model fitting indicators.**

| Model Fit | $\chi^2/df$ | CFI | TLI | SRMR | RMSEA (90%CI) |
|---|---|---|---|---|---|
| Model | 5.122 | 0.983 | 0.977 | 0.024 | 0.040 (0.035-0.045) |

Table 6 provides the model fitting indicators. The structural equation model showed an excellent fit to the data, with $\chi^2/df = 5.122$, CFI = 0.983, TLI = 0.977, SRMR = 0.024, and RMSEA = 0.040 (90% CI [0.035–0.045]). These indices meet the recommended thresholds, indicating that the hypothesized model is well-specified.

Fig 3 illustrates the structural equation model. All path coefficients are statistically significant at the 0.001 level, supporting the relationships among the key variables. Physical self-esteem significantly predicted exercise motivation ($\beta = 0.56$, $p < 0.001$) and social support ($\beta = 0.17$, $p < 0.001$), while exercise motivation ($\beta = 0.56$, $p < 0.001$) and social support ($\beta = 0.28$, $p < 0.001$) significantly predicted sports participation. Additionally, physical self-esteem directly predicted sports participation ($\beta = 0.10$, $p < 0.001$).

Table 7 summarizes the total, direct, and indirect effects. The total effect of physical self-esteem on sports participation was 0.466 ($p < 0.001$), with 20.39% being direct and 79.61% mediated through exercise motivation and social support.

The mediation analysis results are summarized in Table 7:

Direct Effect: Physical self-esteem directly predicted sports participation ($\beta = 0.095$, $p = 0.001$), supporting H1.

Indirect Effects: The mediation effect of exercise motivation ($\beta = 0.155$, $p = 0.001$) accounted for 33.26% of the total effect, supporting H2. The mediation effect of social support ($\beta = 0.078$, $p = 0.001$) accounted for 16.74% of the total effect, supporting H3. The sequential mediation effect of exercise motivation and social support ($\beta = 0.138$, $p = 0.001$) accounted for 29.61% of the total effect, supporting H4.

The findings validate all four hypotheses:

H1: Physical self-esteem significantly and positively predicts adolescents' sports participation.

H2: Exercise motivation mediates the relationship between physical self-esteem and sports participation.

H3: Social support mediates the relationship between physical self-esteem and sports participation.

H4: Exercise motivation and social support sequentially mediate the relationship between physical self-esteem and sports participation.

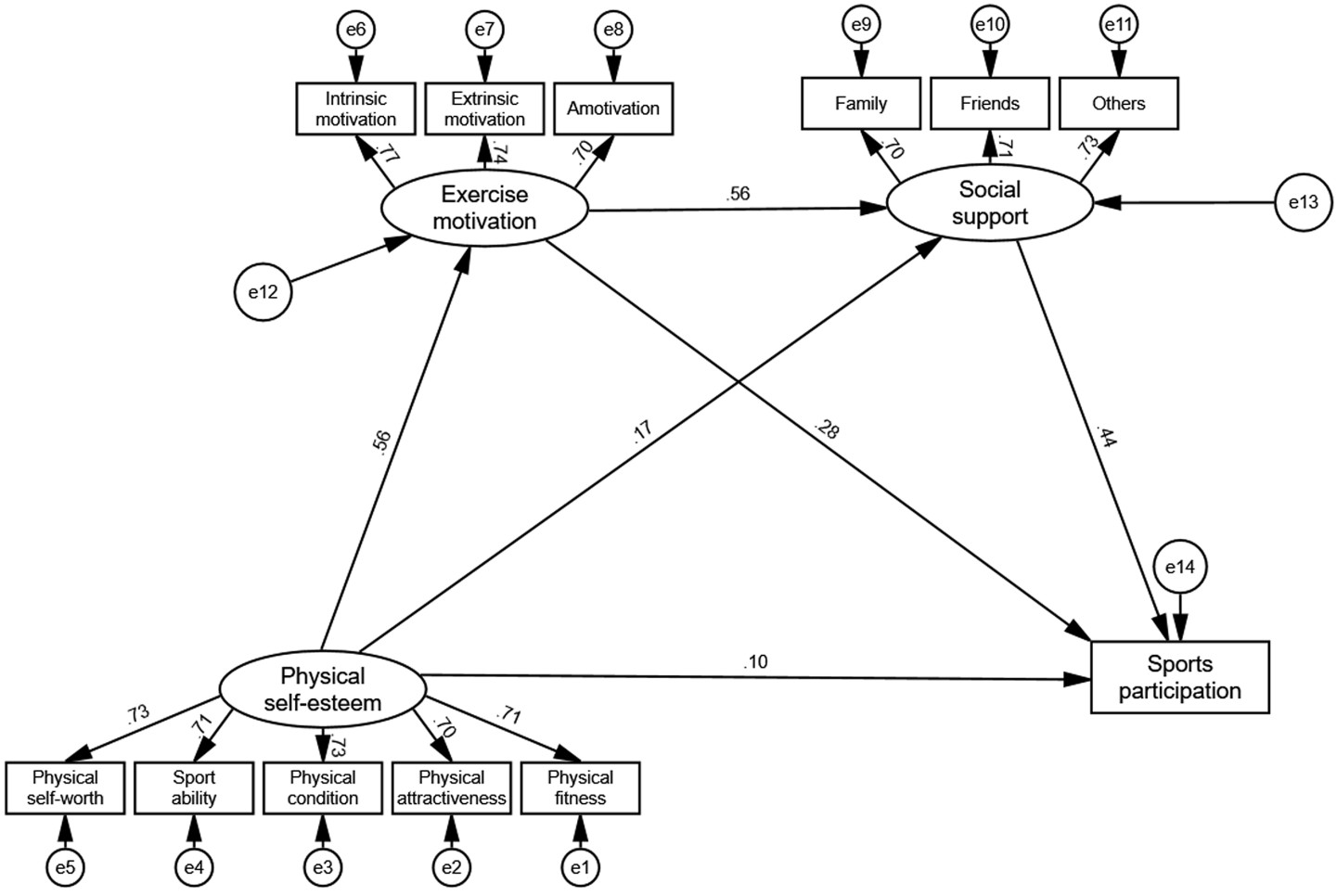

**Fig 3. Structural equation model.** All paths are significant at the 0.001 level.

**Table 7. Total, direct and indirect effects in the multiple mediator model.**

| Path | Estimated effect | Boot SE | P | Boot LLCI | Boot ULCI | Ratio |
|---|---|---|---|---|---|---|
| **Direct effect** | | | | | | |
| PSE→SP | 0.095 | 0.023 | 0.001 | 0.050 | 0.139 | 20.39% |
| **All indirect effects** | | | | | | 79.61% |
| PSE→EM→SP | 0.155 | 0.017 | 0.001 | 0.125 | 0.189 | 33.26% |
| PSE→SS→SP | 0.078 | 0.014 | 0.001 | 0.051 | 0.106 | 16.74% |
| PSE→EM→SS→SP | 0.138 | 0.011 | 0.001 | 0.117 | 0.162 | 29.61% |
| **Total effect** | 0.466 | 0.017 | 0.000 | 0.432 | 0.501 | 100% |

PSE, Physical Self-Esteem; SP, Sports participation; EM, Exercise Motivation; SS, Social Support; Boot LLCI, the lower bound of the 95% confidence interval. Boot ULCI, the upper limit of the 95% confidence interval (Percentile Bootstrap Method with Bias Correction). The Bootstrap sample size is set at 2000.

These results underscore the importance of exercise motivation and social support as mechanisms linking physical self-esteem to sports participation, highlighting the need for interventions targeting these mediators to promote adolescent engagement in sports.

## 5. Discussion

This study provides valuable insights into the mechanisms linking physical self-esteem to sports participation among adolescents. The results validate the hypothesized relationships and emphasize the critical roles of exercise motivation and social support in fostering sports engagement. While we discuss the interrelations among the four key factors, this study's major contribution lies in uncovering the multidimensional impact of sports participation on adolescents, particularly in enhancing psychological, social, and academic outcomes. Below, we delve into the key findings, comparing them to existing literature and exploring their practical implications.

### 5.1. Physical self-esteem as a predictor of sports participation

The results confirm that physical self-esteem significantly and positively predicts adolescents' sports participation (H1). This finding aligns with previous research highlighting the importance of self-perception in motivating sports engagement [6,25]. Adolescents with higher physical self-esteem tend to feel more confident in their physical abilities, which subsequently encourages greater participation in sports. This underscores the importance of interventions aimed at enhancing physical self-esteem, such as skill-building programs and positive reinforcement in sports settings. More importantly, sports participation, as a key outcome, has been shown to improve not only physical fitness but also psychological well-being and social competence among adolescents.

### 5.2. Mediating role of exercise motivation

The mediating role of exercise motivation between physical self-esteem and sports participation (H2) indicates that higher physical self-esteem enhances adolescents' intrinsic or extrinsic motivation to exercise, thereby increasing their level of sports participation. Exercise motivation explained 33.26% of the total effect, underscoring its importance as a mediating pathway. More deeply, sports participation, as a key outcome, is closely related to improvements in psychological health, emotional regulation, and social skills. According to self-determination theory [4], motivation is the core factor driving behavior. Encouraging adolescents to engage in sports not only enhances physical health but also promotes the development of emotional and social skills. Future interventions should focus on cultivating adolescents' intrinsic motivation, particularly by enhancing their motivation to participate through the enjoyment of physical activities, which will further promote their overall development.

### 5.3. Mediating role of social support

Social support also plays a significant mediating role between physical self-esteem and sports participation (H3), explaining 16.74% of the total effect. Sports participation is not only a result of individual factors but is also influenced by the social environment, including family, friends, and peers [16,33,49]. Social support indirectly promotes adolescents' physical self-esteem and exercise motivation by increasing opportunities and confidence to participate in physical activities. Research has shown that social support not only enhances motivation for sports participation but also improves adolescents' emotional health and promotes psychological resilience. To amplify this effect, family-oriented sports activities, peer mentoring, and the establishment of social support networks are particularly important. These measures help increase adolescents' motivation to engage in physical activities, thereby having a profound impact on their psychological development.

### 5.4. Sequential mediation of exercise motivation and social support

The sequential mediation pathway through both exercise motivation and social support (H4) further underscores the interconnected nature of these mediators. This pathway accounted for 29.61% of the total effect, indicating that physical

self-esteem influences sports participation not only through individual motivation but also through the social context. These findings highlight the complexity of adolescent sports behavior and emphasize the need for comprehensive strategies that address both personal and social factors. Sports participation in this context is seen not only as a way to improve physical health but also as an essential factor in enhancing social skills, emotional resilience, and academic achievement.

### 5.5. Demographic differences

This study reveals that gender, grade level, and family location significantly affect adolescents' physical self-esteem, exercise motivation, social support, and sports participation levels. Males tend to score higher than females on all variables, potentially due to societal expectations and cultural reinforcement of male sports participation. High school students exhibit stronger physical self-esteem and exercise motivation, while middle school students tend to rely more on social support. Rural adolescents demonstrate higher participation rates and stronger support networks, likely due to better access to sports resources and stronger community interaction in rural areas. Existing studies often focus on single-variable analyses, neglecting the dynamic relationships between gender, grade level, and urban-rural disparities [14,27]. By integrating a multidimensional model, this study systematically uncovers the comprehensive impacts of demographic variables on sports participation, offering theoretical support for interventions tailored to specific group needs. For instance, interventions targeting female adolescents can focus on enhancing physical self-esteem and intrinsic motivation, while urban adolescents may benefit from community-based activities to strengthen social support.

### 5.6. Implications for practice and policy

The findings of this study provide not only guidance for practical interventions but also valuable insights for policymakers in promoting adolescent sports participation. Firstly, the central role of physical self-esteem in adolescent sports participation highlights the need for a stronger focus on enhancing adolescents' physical self-esteem in practice. Schools and community programs should work towards developing adolescents' confidence in their physical abilities through skill-building, positive feedback, and body-positive campaigns. Targeted interventions for adolescents with lower self-esteem can effectively boost their confidence, thereby motivating them to participate in sports activities.

Secondly, fostering exercise motivation is crucial in promoting adolescent sports participation. Policymakers should prioritize the cultivation of intrinsic motivation, particularly by emphasizing the enjoyment and personal achievement that comes from sports. For example, offering a variety of sports programs can encourage adolescents to engage in physical activities while enhancing their long-term interest and motivation. Moreover, extrinsic motivation, such as rewards and recognition, can also play a role in encouraging sports participation, particularly for short-term engagement.

In terms of social support, this study underscores the significant role that family, peers, and community networks play in promoting adolescent sports participation. Therefore, policies should encourage collaboration among families, schools, and communities to create a supportive environment for adolescents. Governments and educational authorities should actively promote initiatives such as family sports days and peer support groups to provide adolescents with the social support they need to overcome barriers to sports participation. Particularly in regions with significant urban-rural disparities, policies should ensure that all adolescents have access to adequate social support and resources, which would help achieve higher participation rates.

Lastly, this study reveals that adolescent sports participation is significantly influenced by factors such as gender, grade level, and geographic location. In response, policies should adopt more inclusive and targeted measures, especially gender-sensitive strategies, to reduce participation gaps among different groups. For example, interventions aimed at female adolescents can focus on enhancing physical self-esteem and fostering intrinsic motivation, while urban adolescents may benefit more from community-based activities that strengthen their social support networks.

In conclusion, policymakers should consider the multifaceted factors influencing adolescent sports participation, including individual, family, and social support. Policies should not only focus on providing sports resources and opportunities but also promote the psychological and emotional development of adolescents to ensure comprehensive support for their participation in sports activities. Through these multi-level policy interventions, we can create a more inclusive and supportive environment for adolescent sports participation, fostering their overall physical and mental development.

## 6. Limitations and future directions

Although this study provides valuable insights, there are several limitations that warrant consideration. First, the cross-sectional design limits the ability to infer causal relationships. Future longitudinal studies are needed to investigate the temporal dynamics among physical self-esteem, exercise motivation, social support, and sports participation, which would allow for a more robust understanding of causal pathways and developmental patterns over time.

Second, the study relied entirely on self-reported data, which may be subject to biases such as social desirability or recall errors. To address this, future research could integrate objective measures—such as wearable devices (e.g., fitness trackers, smartwatches) or behavioral observation protocols—to assess adolescents' actual physical activity levels in terms of frequency, duration, and intensity. Such methods would help to validate self-report data and improve the accuracy of behavioral assessments.

Third, while the current study conceptualized social support as a mediating variable, its contextual and external nature also lends itself to a potential moderating role. Future research could explore moderated mediation models to examine whether social support strengthens or attenuates the pathways from physical self-esteem to sports participation via motivation. This would provide a more nuanced understanding of how individual and environmental factors interact to shape adolescent behavior.

Fourth, although the structural model was tested across a large and diverse sample, measurement invariance across key demographic subgroups—such as school stage and residence type—was not assessed in this study. Establishing measurement and structural invariance in future work will be essential to ensure that the proposed relationships hold consistently across different population segments and contextual settings.

Finally, additional mediators and mechanisms remain to be explored. Psychological health indicators (e.g., anxiety, depressive symptoms) and environmental resources (e.g., family cohesion, school-based sports facilities) may also play important roles in the interplay between self-perception and sports behavior. Incorporating these variables in future models would enable researchers to construct a more comprehensive and ecologically valid framework for understanding adolescent sports participation and its broader psychosocial outcomes.

## 7. Conclusion

In conclusion, this study demonstrates that physical self-esteem significantly influences adolescents' sports participation, both directly and indirectly through exercise motivation and social support. The findings highlight the importance of fostering intrinsic motivation and creating supportive social environments to enhance sports engagement among adolescents. Furthermore, the significant demographic differences emphasize the need for tailored and inclusive interventions. By addressing both individual and social factors, these insights can inform policies and programs aimed at promoting physical activity and overall well-being in adolescents.

## Supporting information

**S1 Data. Empirical data and original scale.**
(ZIP)

## Acknowledgments

The authors thank all individuals who participated in this study.

## Author contributions

**Data curation:** Weisong Chen, Hongshen Wang.

**Formal analysis:** Bo Peng, Weisong Chen.

**Investigation:** Bo Peng, Weisong Chen, Ting Yu.

**Methodology:** Bo Peng, Ting Yu.

**Project administration:** Hongshen Wang.

**Writing – original draft:** Bo Peng, Weisong Chen, Hongshen Wang, Ting Yu.

**Writing – review & editing:** Bo Peng, Weisong Chen, Hongshen Wang, Ting Yu.

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
