## [Decision Letter · Decision Letter 0]

12 Jan 2025

PONE-D-24-57757From physical self-esteem to sports participation: the mediating role of exercise motivation and social support in adolescentsPLOS ONE

Dear Dr. wang,

Thank you for submitting your manuscript to PLOS ONE. After careful consideration, we feel that it has merit but does not fully meet PLOS ONE’s publication criteria as it currently stands. Therefore, we invite you to submit a revised version of the manuscript that addresses the points raised during the review process.

**ACADEMIC EDITOR: **

Dear authorThe article you submitted is interesting and has been sent to the reviewers for comments and opinions.After reading the reviewers' comments, it was decided to recommend that you address each of the comments, and resubmit the article. Pay special attention to reviewer 2's comments.In addition we would like to see a reference to each of the comments in an attached letter.

We look forward to receiving your revised manuscript.

Kind regards,

Gal Harpaz, Ph.D.

Academic Editor

PLOS ONE

Journal Requirements:

3. In the online submission form, you indicated that the original contributions presented in the study are included in the article/supplementary material, further inquiries can be directed to the corresponding author/s. 

Reviewers' comments:

Reviewer's Responses to Questions

**Comments to the Author**

1. Is the manuscript technically sound, and do the data support the conclusions?

Reviewer #1: Yes

Reviewer #2: Yes

2. Has the statistical analysis been performed appropriately and rigorously? 

Reviewer #1: Yes

Reviewer #2: Yes

3. Have the authors made all data underlying the findings in their manuscript fully available?

Reviewer #1: Yes

Reviewer #2: No

4. Is the manuscript presented in an intelligible fashion and written in standard English?

Reviewer #1: Yes

Reviewer #2: Yes

5. Review Comments to the Author

Reviewer #1: Dear Authors,

Congratulations of a very intersesting and well written manuscript.

In my opinion the manuscript follows the Journal guidelines and is well prepared for publication with a need of minor amendements especially in:

1. Tables 2,4, and 8 - please double check and remove "squares" and replace with the correct one in the tables indicated.

2. Please also double check the text for punctuation mistakes.

3. In point 6 Limitations and future directions, please elaborate what kind of the objective measures of sports participation the Authors have on their mind, to give a clear directions for future research, and demonstrate a solid understanding of the focused field.

Kind regards,

Reviewer

Reviewer #2: The manuscript presents an interesting study examining the links between physical self-esteem, exercise motivation, social support, and sports participation. The authors use advanced structural equation modelling to explore the connections and also test for the moderating role of gender. The advantages of the study include the large sample size, the innovative conceptual framework, ethical execution of the study, and the advanced analytic techniques. Please find below my feedback for improving the manuscript.

The argument about the importance of sports participation, as the key outcome, should be beefed up. Now, the authors spend more time on the interrelations between the four factors of interest but do not stress enough the importance of sports participation for various adolescent outcomes.

The introduction (page 9) states “research on how these three factors (physical self-esteem, exercise motivation, and social support) interact and jointly influence…”. The term “interact” is inappropriate here since it refers to statistical interaction which is not the analytic focus here.

The authors introduce physical self-esteem without introducing the definition of this factor. Also, I am wondering whether there is a need to introduce self-esteem as a broader construct before specialising further.

Similarly, exercise motivation is presented as a broad concept. Yet, motivation is not a unitary concept; there are several theories of motivation (e.g., goals, intrinsic motivation vs. extrinsic, value). Latter on, the authors present the concept of intrinsic motivation in that particular section. I would advise to refine and define the concept before delving deeper into the relevant literature.

I find it difficult to locate all the hypotheses interspersed inside the literature review. Please consider transferring all hypotheses to a ‘present study’/ ‘theoretical model’ section.

Regarding power analysis, I am afraid that I fail to see the relevance of power analysis here considering that power analysis was done only for multiple regression, whereas the authors used SEM modelling. SEM modelling power analysis requires simulation.

There is a mention of multi-stage cluster sampling, yet it is unclear whether this kind of sampling design influences the variables under study. Perhaps, it makes sense here to report the intra-class correlation coefficient per key variable to understand the potential impact of ignoring the cluster sampling.

In the ‘data analysis procedures’, please add a citation to the acceptable levels for the fit indices. Furthermore, explain the criteria for testing measurement invariance (the levels of invariance tested and the fit indices utilised to assess the (non-)invariance).

In table 1, presenting cumulative percentage does not add anything more compared to percentage.

Harman’s single factor test for CMB has been criticised in the literature. I would suggest complimenting Harman’s test with another alternative test via SEM (see the Unmeasured latent method factor).

Since the fit indices are presented in table 3, I find it a loss of space to replicate the fit indices inside the flowing text.

The authors present t-tests for testing group differences between gender, grade, and family location. However, it is known that strong measurement invariance is required before directly comparing the means of observed variables (see Kline, 2023). So, I advise testing for invariance before comparing, though I am wondering why it makes sense to compare here since this is not the key focus of the study.

Inside the results’ section, I encountered a test of structural invariance across gender groups. However, the authors have not presented any theoretical evidence suggesting that gender potentially moderates the relationships between these four key factors. Demographic differences are also not of key focus here and the authors have not explained how demographic factors influence variations in the four key variables under study.

Section 5.7 appears as bullet points. I advise restructuring the text into flowing text. In that specific section, I would like to see a greater emphasis on policy implications because now it is entirely focused on practical implications.

I hope the authors will find my comments helpful in improving their arguments and their approach!

Kind regards,

The reviewer

6. PLOS authors have the option to publish the peer review history of their article (what does this mean? ). If published, this will include your full peer review and any attached files.

**Do you want your identity to be public for this peer review?** For information about this choice, including consent withdrawal, please see our Privacy Policy .

Reviewer #1: No

Reviewer #2: **Yes: ** Dr Ioannis G. Katsantonis

---

## [Author Response · Author response to Decision Letter 1]

1 Mar 2025

Response to Reviewers

We would like to express our sincere gratitude to both reviewers for their thorough and constructive comments. Your insightful feedback has been invaluable in helping us improve the quality, rigor, and clarity of our manuscript. We carefully reviewed all the comments and have made substantial revisions to address the concerns raised. Below, we provide a point-by-point response to each of the reviewers’ comments, detailing the revisions we have made.

Once again, we are deeply grateful for your time, expertise, and constructive suggestions, which have significantly strengthened our study. We hope that the revised manuscript meets your expectations.

Reviewer 1

Comment 1:Please check tables 2, 4, and 8.

Reviewer’s Comment:Tables 2,4, and 8 - please double check and remove "squares" and replace with the correct one in the tables indicated.

Response:Thank you for your valuable suggestions and thorough evaluation of our research. Your feedback has helped us critically reflect on the issues in the study and provided clear directions for further improvements. In response to your suggestion, we have carefully reviewed Tables 2, 4, and 8 and have replaced the incorrect use of "squares" with the correct terminology to ensure accuracy and adherence to academic standards.

Revisions Made:

Table 2. Scales used in this study.

Scale Author (Year) Item quantity Scoring Dimensions

Physical self-esteem Xia Xu(2001) 30 4 Physical self-worth;

Sport ability;

Physical condition;

Physical attractiveness;

Physical fitness

Physical exercise Deqing Liang (1994) 3 5 Exercise intensit; Duratio;

exercise Frequency

Exercise motivation Liwei Zhang

2004� 28 7 Intrinsic motivation;

Extrinsic motivation;

Amotivation

Social support Wei Chen(2016) 12 5 Family;

Friends;

Others

Table 4. Comparative fit indices for alternative factor structures of key constructs.

Model Factor χ 2 df Delta χ2

(Delta df) CFI TLI SRMR RMSEA (90%CI)

Four-factor model PSE,EM,SS,SP 363.16 71 - 0.980 0.974 0.025 0.040 (0.036-0.044)

Three-factor model PSE+EM,SS,SP 2002.29 74 -0.113

(-0.137) 0.867 0.837 0.078 0.100 (0.097-0.104)

Two-factor model PSE+EM+SS,SP 3186.75 76 -0.194

(-0.231) 0.786 0.743 0.089 0.126 (0.122-0.130)

One-factor model PSE+EM+SS+SP 3722.29 77 -0.231

(-0.271) 0.749 0.703 0.093 0.135 (0.132-0.139)

PSE,Physical Self-Esteem. SP, Sports participation. EM, Exercise Motivation. SS,Social Support.All △χ 2 passed the significance test at 0.01 level.

Table 8. Testing for structural invariance across gender.

χ 2/df CFI Delta CFI TLI Delta

TLI SRMR RMSEA

(90%CI)

Unconstrained 3.079 0.982 - 0.976 - 0.029 0.028 (0.025-0.032)

Measurement weights 3.128 0.980 -0.002 0.975 -0.001 0.030 0.029 (0.025-0.032)

Structural weights 3.092 0.980 -0.002 0.976 0.000 0.032 0.028 (0.025-0.032)

Structural covariances 3.070 0.980 -0.002 0.976 0.000 0.032 0.028 (0.025-0.032)

Structural residuals 3.023 0.980 -0.002 0.977 +0.001 0.032 0.028 (0.025-0.031)

Specific Modifications:

We carefully reviewed Tables 2, 4, and 8, and corrected the incorrect symbols in the tables to ensure that all terms are accurate and consistent, in accordance with academic standards.

Comment 2:Check punctuation marks.

Reviewer’s Comment:Please also double check the text for punctuation mistakes.

Response:Thank you for your valuable suggestions and in-depth evaluation of our research. Your feedback has helped us critically reflect on the issues in our study and provided clear directions for further improvement. In response to your suggestion, we have carefully checked the punctuation throughout the manuscript and made corrections to any inappropriate punctuation to ensure accuracy and compliance with academic standards.

Revisions Made:

1 Introduction

Specific Modifications:

Removal of the Colon (":") after "1 Introduction":The original text included the colon after the title "1 Introduction:".The colon has been removed to adhere to academic standards and formatting guidelines for section headings. In most academic writing styles, section titles do not require a colon unless they are followed by a subtitle. Since there is no subtitle following "Introduction," the colon is unnecessary.This adjustment improves the presentation of the section heading and aligns it with proper academic formatting conventions.

Comment 3:Limitations and future directions.

Reviewer’s Comment:In point 6 Limitations and future directions, please elaborate what kind of the objective measures of sports participation the Authors have on their mind, to give a clear directions for future research, and demonstrate a solid understanding of the focused field.

Response:Thank you for your valuable suggestions and in-depth evaluation of our study. Your feedback has prompted us to critically reflect on the issues in our research and provided clear directions for further improvement. In response to your suggestions, we have elaborated in Section 6 "Limitations and Future Directions" on the objective measures that can be adopted in future research to better assess sports participation. We have explicitly proposed the use of activity trackers, wearable devices (such as smartwatches), and behavioral observations to collect more accurate data, thereby enhancing the reliability of the study.

Revisions Made:

6 Limitations and future directions

Although this study provides valuable insights, there are some limitations that should be acknowledged. First, the cross-sectional design of this study limits the ability to infer causal relationships. Future longitudinal studies could more deeply explore the dynamic relationships between these variables over time, allowing for a more accurate understanding of causal pathways and long-term effects.

Second, this study relied on self-reported measures, which may introduce biases such as social desirability bias or recall bias. To reduce these biases, future research could incorporate objective measures of sports participation, such as activity trackers, wearable devices (e.g., smartwatches), or behavioral observations to obtain more precise data. These tools could provide multidimensional information on the frequency, duration, and intensity of adolescent physical activity, enhancing the accuracy and reliability of the data.

Additionally, future research could further explore other potential mediators, particularly psychological health factors (e.g., anxiety, depression symptoms) and environmental factors (e.g., family support, school resources). By incorporating these factors, researchers can gain a more comprehensive understanding of how physical exercise influences adolescents' psychological resilience and self-efficacy through multiple mechanisms, providing theoretical support for more effective intervention strategies.

Specific Modifications:

1:Clearer objective measurement tools: We have mentioned that future research should use objective measurement tools such as activity trackers, wearable devices (e.g., smartwatches), and behavioral observations. These tools can more accurately record and analyze various parameters of physical activity.

2:A deeper exploration of future research directions: We have expanded the discussion on potential mediators to be included in future research, particularly the roles of psychological health and environmental factors. This will help provide a more comprehensive understanding of the relationship between physical exercise and psychological development.

3:Strengthened recommendations for causal inference: By proposing longitudinal studies, we have clearly articulated the need to explore the dynamic changes in causal relationships, enhancing the depth and feasibility of future research.

We sincerely appreciate your insightful comments, which have greatly enhanced the clarity, accuracy, and overall presentation of the manuscript. Thank you for your valuable feedback.

Reviewer 2

Comment 1:Advantages of the manuscript.

Reviewer’s Comment:The manuscript presents an interesting study examining the links between physical self-esteem, exercise motivation, social support, and sports participation. The authors use advanced structural equation modelling to explore the connections and also test for the moderating role of gender. The advantages of the study include the large sample size, the innovative conceptual framework, ethical execution of the study, and the advanced analytic techniques.

Response:We sincerely appreciate the reviewer’s thoughtful and constructive feedback. Thank you for recognizing the strengths of our study, including the large sample size, the innovative conceptual framework, the ethical execution of the study, and the advanced analytic techniques employed. Your positive comments on the use of structural equation modeling and the exploration of the moderating role of gender are greatly valued. We are grateful for your detailed review, which has not only highlighted the merits of our work but also provided us with important perspectives for enhancing the quality and depth of our research. Your input has been invaluable in guiding the further refinement of our manuscript.

Comment 2:The argument about the importance of sports participation, as the key outcome, should be beefed up.

Reviewer’s Comment:The argument about the importance of sports participation, as the key outcome, should be beefed up. Now, the authors spend more time on the interrelations between the four factors of interest but do not stress enough the importance of sports participation for various adolescent outcomes.

Response:Thank you for your valuable comments and in-depth evaluation of our study. Your feedback has prompted us to reflect more deeply on the focus of our research and has enhanced our understanding of the importance of sports participation. In response to your suggestions, we have made comprehensive revisions to the relevant content, further strengthening the discussion of sports participation as a key outcome, and emphasizing its critical role in adolescent mental health, emotional regulation, and social skills. We believe that, with these adjustments, the discussion section now more effectively balances the interrelationships among the four factors and highlights the profound impact of sports participation on various aspects of adolescent development.

Revisions Made:

5 Discussion

This study provides valuable insights into the mechanisms linking physical self-esteem to sports participation among adolescents. The results validate the hypothesized relationships and emphasize the critical roles of exercise motivation and social support in fostering sports engagement. While we discuss the interrelations among the four key factors, this study’s major contribution lies in uncovering the multidimensional impact of sports participation on adolescents, particularly in enhancing psychological, social, and academic outcomes. Below, we delve into the key findings, comparing them to existing literature and exploring their practical implications.

5.1 Physical self-esteem as a predictor of sports participation

The results confirm that physical self-esteem significantly and positively predicts adolescents’ sports participation (H1). This finding aligns with previous research highlighting the importance of self-perception in motivating sports engagement [6, 25]. Adolescents with higher physical self-esteem tend to feel more confident in their physical abilities, which subsequently encourages greater participation in sports. This underscores the importance of interventions aimed at enhancing physical self-esteem, such as skill-building programs and positive reinforcement in sports settings. More importantly, sports participation, as a key outcome, has been shown to improve not only physical fitness but also psychological well-being and social competence among adolescents.

5.2 Mediating role of exercise motivation

The mediating role of exercise motivation between physical self-esteem and sports participation (H2) indicates that higher physical self-esteem enhances adolescents' intrinsic or extrinsic motivation to exercise, thereby increasing their level of sports participation. Exercise motivation explained 33.26% of the total effect, underscoring its importance as a mediating pathway. More deeply, sports participation, as a key outcome, is closely related to improvements in psychological health, emotional regulation, and social skills. According to self-determination theory [4], motivation is the core factor driving behavior. Encouraging adolescents to engage in sports not only enhances physical health but also promotes the development of emotional and social skills. Future interventions should focus on cultivating adolescents' intrinsic motivation, particularly by enhancing their motivation to participate through the enjoyment of physical activities, which will further promote their overall development.

5.3 Mediating role of social support

Social support also plays a significant mediating role between physical self-esteem and sports participation (H3), explaining 16.74% of the total effect. Sports participation is not only a result of individual factors but is also influenced by the social environment, including family, friends, and peers [16,33,49]. Social support indirectly promotes adolescents' physical self-esteem and exercise motivation by increasing opportunities and confidence to participate in physical activities. Research has shown that social support not only enhances motivation for sports participation but also improves adolescents' emotional health and promotes psychological resilience. To amplify this effect, family-oriented sports activities, peer mentoring, and the establishment of social support networks are particularly important. These measures help increase adolescents' motivation to engage in physical activities, thereby having a profound impact on their psychological development.

5.4 Sequential mediation of exercise motivation and social support

The sequential mediation pathway through both exercise motivation and social support (H4) further underscores the interconnected nature of these mediators. This pathway accounted for 29.61% of the total effect, indicating that physical self-esteem influences sports participation not only through individual motivation but also through the social context. These findings highlight the complexity of adolescent sports behavior and emphasize the need for comprehensive strategies that address both personal and social factors. Sports participation in this context is seen not only as a way to improve physical health but also as an essential factor in enhancing social skills, emotional resilience, and academic achievement.

5.5 Demographic differences

This study reveals that gender, grade level, and family location significantly affect adolescents' physical self-esteem, exercise motivation, social support, and sports participation levels. Males tend to score higher than females on all variables, potentially due to societal expectations and cultural reinforcement of male sports participation. High school students exhibit stronger physical self-esteem and exercise motivation, while middle school students tend to rely more on social support. Rural adolescents demonstrate higher participation rates and stronger support networks, likely due to better access to sports resources and stronger community interaction in rural areas. Existing studies often focus on single-variable analyses, neglecting the dynamic relationships between gender, grade level, and urban-rural disparities [14, 27]. By integrating a multidimensional model, this study systematically uncovers the comprehensive impacts of demographic variables on sports participation, offering theoretical support for interventions tailored to specific group needs. For instance, interventions targeting female adolescents can focus on enhancing physical self-esteem and intrinsic motivation, while urban adolescents may benefit from community-based activities to strengthen social support.

5.6 Implications for practice and policy

The findings of this study provide not only guidance for practical interventions but also valuable insights for policymakers in promoting adolescent sports participation. Firstly, the central role of physical self-est

---

## [Decision Letter · Decision Letter 1]

6 Apr 2025

PONE-D-24-57757R1From physical self-esteem to sports participation: the mediating role of exercise motivation and social support in adolescentsPLOS ONE

Dear Dr. wang,

Thank you for submitting your manuscript to PLOS ONE. After careful consideration, we feel that it has merit but does not fully meet PLOS ONE’s publication criteria as it currently stands. Therefore, we invite you to submit a revised version of the manuscript that addresses the points raised during the review process.

Thank you for submitting the revised article. Attention to detail and reviewer comments on the original version is evident.

The revised version has been sent to three reviewers. Please see the reviewers' few comments and submit a revised version.

We look forward to receiving your revised manuscript.

Kind regards,

Gal Harpaz, Ph.D.

Academic Editor

PLOS ONE

Journal Requirements:

Reviewers' comments:

Reviewer's Responses to Questions

**Comments to the Author**

1. If the authors have adequately addressed your comments raised in a previous round of review and you feel that this manuscript is now acceptable for publication, you may indicate that here to bypass the “Comments to the Author” section, enter your conflict of interest statement in the “Confidential to Editor” section, and submit your "Accept" recommendation.

Reviewer #1: All comments have been addressed

Reviewer #3: All comments have been addressed

Reviewer #4: All comments have been addressed

2. Is the manuscript technically sound, and do the data support the conclusions?

Reviewer #1: Yes

Reviewer #3: Yes

Reviewer #4: Yes

3. Has the statistical analysis been performed appropriately and rigorously? 

Reviewer #1: Yes

Reviewer #3: Yes

Reviewer #4: Yes

4. Have the authors made all data underlying the findings in their manuscript fully available?

Reviewer #1: Yes

Reviewer #3: Yes

Reviewer #4: Yes

5. Is the manuscript presented in an intelligible fashion and written in standard English?

Reviewer #1: Yes

Reviewer #3: Yes

Reviewer #4: Yes

6. Review Comments to the Author

Reviewer #1: The Authors adressed all recommended amendments. I reccomend the manuscript to be published in PLOS ONE.

Reviewer #3: I recently had the opportunity to read Author´s manuscript. It shows a noteworthy study with sophisticated analysis. I would like to highlight that the Authors dedicated time and effort to address all comments made by the previous Reviewers. I compared the original submission to the revised one and, the modifications improved substantially the structure and the content of the manuscript. I appreciate the endeavor and creativity the Authors invested in this work.

I also would like to raise some aspects that could be improved:

Comment 1: Figure 1 is mentioned in Theoretical Model, however the Figure 1 per se is not available throughout the document.

Comment 2: I would word the Hypothesis4(H4) to emphasize the joint mediation that occurs with exercise motivation and social support: “Exercise motivation and social support jointly mediate the relationship between physical self-esteem and sports participation in adolescents.” For reference: VanderWeele TJ, Vansteelandt S. Mediation Analysis with Multiple Mediators. Epidemiol Methods. 2014 Jan;2(1):95-115. doi: 10.1515/em-2012-0010. PMID: 25580377; PMCID: PMC4287269.

Comment 3: I understood that Table 8 “Testing for structural invariance across gender” was removed to address previous Reviewer’s request. Yet, the following sentence was found: “The mediation analysis should be summarized in Table 8.” I suggest including the new Table 8 to the manuscript, or removing the sentence.

Comment 4: I noticed that the Authors corrected the squares that appeared in Table 2, as previous recommendations. However, parenthesis observed in Liwei Zhang (2004), found in row “Exercise motivation”, column “Author (Year)”, seem different from the others (perhaps a

different font? Size? Space?).

Reviewer #4: From the manuscript as is currently written, I think the authors might want to investigate the following: the role of social support as a moderator, which would then be a moderated mediation analysis. Because social support is extrinsic to an individual, this is why it might be more of a moderator than a mediator. Social support is outside of the control of an individual, so it moderates the relation between your independent and dependent variable, along with the mediating role of exercise motivation.

Further, although in the study, it was found that the structure of the model was invariant across gender, additional testing for invariance should have been conducted for school stage and residence type.

7. PLOS authors have the option to publish the peer review history of their article (what does this mean? ). If published, this will include your full peer review and any attached files.

**Do you want your identity to be public for this peer review?** For information about this choice, including consent withdrawal, please see our Privacy Policy .

Reviewer #1: No

Reviewer #3: No

Reviewer #4: No

---

## [Author Response · Author response to Decision Letter 2]

21 Apr 2025

Response to Reviewers

We would like to express our sincere gratitude to all three reviewers for their thorough and constructive comments on our manuscript. Your insightful feedback has been invaluable in helping us improve the quality, rigor, and clarity of our manuscript. We carefully reviewed all the comments and have made substantial revisions to address the concerns raised. Below, we provide a point-by-point response to each of the reviewers’ comments, detailing the revisions we have made.

Once again, we deeply appreciate your time, expertise, and constructive suggestions, which have significantly strengthened our study. We hope that the revised manuscript meets your expectations.

Reviewer 1

Comment 1:I reccomend the manuscript to be published in PLOS ONE.

Reviewer’s Comment:The Authors adressed all recommended amendments. I reccomend the manuscript to be published in PLOS ONE.

Response:Thank you to the reviewer for recognizing and supporting our manuscript. We greatly appreciate your acknowledgment of the revisions we made and for recommending our manuscript for publication in PLOS ONE. Your valuable feedback and guidance have played a crucial role in improving the quality of our research.

Reviewer 3

Comment 1:The recognition of the research.

Reviewer’s Comment: I recently had the opportunity to read Author´s manuscript. It shows a noteworthy study with sophisticated analysis. I would like to highlight that the Authors dedicated time and effort to address all comments made by the previous Reviewers. I compared the original submission to the revised one and, the modifications improved substantially the structure and the content of the manuscript. I appreciate the endeavor and creativity the Authors invested in this work.

Response:Thank you to the reviewer for the recognition and appreciation of our manuscript. We greatly appreciate your acknowledgment of the time and effort we invested in addressing the comments from previous reviewers. Your praise is highly meaningful to us and motivates us to continue improving the quality of our research. Thank you for recognizing the creativity and effort we put into this work.

Comment 2:Figure 1.

Reviewer’s Comment: Figure 1 is mentioned in Theoretical Model, however the Figure 1 per se is not available throughout the document.

Response:Thank you for your thoughtful feedback. We have noted that Figure 1 is mentioned in the "Theoretical Model" but was not included in the document. We have made the necessary revisions to ensure that Figure 1 is properly presented in the manuscript. Thank you for your valuable suggestion.

Revisions Made:

Figure 1. Theoretical model of the impact of adolescent physical self-esteem on sports participation: the chain mediation effect of exercise motivation and social support.

Specific Modifications:

Thank you for your feedback. In response to your comment about Figure 1 being mentioned in the "Theoretical Model" but not presented in the document, we have now inserted Figure 1 into the manuscript to ensure it is properly displayed in the relevant section. Thank you for your valuable suggestion.

Comment 3:Reformulate the statement of Hypothesis H4.

Reviewer’s Comment:I would word the Hypothesis4(H4) to emphasize the joint mediation that occurs with exercise motivation and social support: “Exercise motivation and social support jointly mediate the relationship between physical self-esteem and sports participation in adolescents.” For reference: VanderWeele TJ, Vansteelandt S. Mediation Analysis with Multiple Mediators. Epidemiol Methods. 2014 Jan;2(1):95-115. doi: 10.1515/em-2012-0010. PMID: 25580377; PMCID: PMC4287269.

Response:Thank you for your valuable suggestion. We have revised Hypothesis H4 based on your feedback to emphasize the joint mediation of exercise motivation and social support. The revised statement is: "Exercise motivation and social support jointly mediate the relationship between physical self-esteem and sports participation in adolescents." Thank you for your careful guidance, which has helped us express the hypothesis more clearly.

Revisions Made:

Hypothesis4(H4): Exercise motivation and social support jointly mediate the relationship between physical self-esteem and sports participation in adolescents.

Specific Modifications:

Thank you for your suggestion. Based on your feedback, we have revised the statement of Hypothesis H4 to better emphasize the joint mediation of exercise motivation and social support. The revised statement is: "Exercise motivation and social support jointly mediate the relationship between physical self-esteem and sports participation in adolescents." This modification makes the hypothesis clearer and aligns with your suggestion. Thank you for your careful guidance.

Comment 4: I suggest including the new Table 8 to the manuscript, or removing the sentence.

Reviewer’s Comment: I understood that Table 8 “Testing for structural invariance across gender” was removed to address previous Reviewer’s request. Yet, the following sentence was found: “The mediation analysis should be summarized in Table 8.” I suggest including the new Table 8 to the manuscript, or removing the sentence.

Response:Thank you for your valuable feedback. We noticed that during the revision process, we mistakenly labeled Table 7 as Table 8. In response to your suggestion, we have made the necessary corrections and revised the inaccurate sentence. Thank you for your careful guidance, which has helped us improve the accuracy of the manuscript.

Revisions Made:

The mediation analysis results are summarized in Table 7:

Specific Modifications:

Thank you for your suggestion. During the revision process, we indeed removed Table 8 "Testing for structural invariance across gender" to address the previous reviewer's request. We noticed that, during the revision, we mistakenly labeled Table 7 as Table 8. In response to your suggestion, we have made the necessary corrections and revised the inaccurate sentence to ensure consistency and accuracy in the manuscript. Thank you for your careful guidance.

Comment 5: The box issue in Table 2.

Reviewer’s Comment: I noticed that the Authors corrected the squares that appeared in Table 2, as previous recommendations. However, parenthesis observed in Liwei Zhang (2004), found in row “Exercise motivation”, column “Author (Year)”, seem different from the others (perhaps a different font? Size? Space?).

Response:Thank you for your detailed feedback. We noticed that the parentheses in Liwei Zhang (2004) in the "Exercise motivation" row and the "Author (Year)" column of Table 2 were different from the others. We have made the necessary corrections to standardize the font, size, and spacing of the parentheses to ensure consistency in the table. Thank you for your valuable suggestion.

Revisions Made:

Table 2. Scales used in this study.

Scale Author (Year) Item quantity Scoring Dimensions

Physical self-esteem Xia Xu(2001) 30 4 Physical self-worth;

Sport ability;

Physical condition;

Physical attractiveness;

Physical fitness

Physical exercise Deqing Liang (1994) 3 5 Exercise intensit; Duratio;

exercise Frequency

Exercise motivation Liwei Zhang

(2004) 28 7 Intrinsic motivation;

Extrinsic motivation;

Amotivation

Social support Wei Chen(2016) 12 5 Family;

Friends;

Others

Specific Modifications:

Thank you for your detailed feedback. We have noticed that the parentheses used for Liwei Zhang (2004) in the "Exercise motivation" row and the "Author (Year)" column of Table 2 were different from the others. To address this, we have standardized the font, size, and spacing of the parentheses to ensure consistency with the rest. Thank you for your valuable suggestion, which has helped improve the consistency of the table.

We sincerely appreciate your insightful comments, which have greatly enhanced the clarity, accuracy, and overall presentation of the manuscript. Thank you for your valuable feedback.

Reviewer 4

Comment 1: The moderating role of social support.

Reviewer’s Comment: From the manuscript as is currently written, I think the authors might want to investigate the following: the role of social support as a moderator, which would then be a moderated mediation analysis. Because social support is extrinsic to an individual, this is why it might be more of a moderator than a mediator. Social support is outside of the control of an individual, so it moderates the relation between your independent and dependent variable, along with the mediating role of exercise motivation.

Response:Thank you very much for this thoughtful and theoretically grounded suggestion. We fully acknowledge that social support, as a contextual and externally sourced factor, is often conceptualized as a moderator in behavioral models, particularly when it is positioned as a buffer or amplifier of the relationship between personal traits and behavioral outcomes.

However, in the present study, we selected a mediating role for social support based on two key considerations:

Perceived social support—the construct used in this study—is not merely an objective external condition, but a subjective experience shaped by personal traits such as self-esteem, emotional openness, and interpersonal style. Prior studies have shown that individuals with higher physical self-esteem are more likely to perceive, seek, and internalize support from others. In this sense, perceived social support may function as a psychosocial response elicited by physical self-esteem rather than a purely exogenous moderator.

Our empirical results show that physical self-esteem significantly predicts perceived social support (β = 0.17, p < 0.001), which supports the logic of a causal path from self-esteem to support perception. Moreover, in our sequential mediation analysis, social support was found to partially transmit the effect of exercise motivation to sports participation, suggesting its functional position within an internal psychological chain rather than as a conditional boundary factor.

That said, we agree with you that an alternative conceptualization of social support as a moderator (e.g., moderating the path from motivation to behavior) is theoretically meaningful and offers a valuable avenue for future research. We have added a corresponding discussion in the Limitations and Future Directions section (Section 6), where we explicitly propose testing a moderated mediation model in future studies. This would allow for a more comprehensive examination of the boundary conditions under which physical self-esteem and motivation influence behavioral engagement.

We appreciate your input, which has helped us further reflect on the conceptual positioning of key variables and enriched the theoretical scope of the study.

Comment 2: Robustness testing.

Reviewer’s Comment:  Further, although in the study, it was found that the structure of the model was invariant across gender, additional testing for invariance should have been conducted for school stage and residence type.

Response:We appreciate your thoughtful suggestion regarding additional invariance testing across school stage and residence type. Indeed, multi-group comparisons can provide valuable insights into the stability of structural relationships across subpopulations.

However, following a recommendation from another reviewer who expressed concerns about the manuscript's length and model complexity, we have removed all measurement invariance testing from the revised version to maintain focus and coherence. This decision was made to streamline the presentation and emphasize the core mediation model and its practical implications.

That said, we fully agree that testing for structural equivalence across demographic groups is a meaningful direction for future research. Accordingly, we have added a note in the Limitations and Future Directions section (Section 6) indicating that future studies should incorporate measurement invariance analyses across different demographic strata (e.g., school stage and residence type) to enhance generalizability and model robustness.

We sincerely appreciate your insightful comments, which have greatly enhanced the clarity, accuracy, and overall presentation of the manuscript. Thank you for your valuable feedback.

Response Summary and Appreciation

We sincerely thank the three reviewers and the editor for their detailed and constructive feedback. We have carefully considered each comment and made substantial revisions to the manuscript to address all concerns. These revisions have enhanced the practicality of the study, the clarity of the discussion, and the actionable nature of the future directions based on the study's limitations. We hope that the revised manuscript meets your expectations and look forward to your further evaluation. Once again, thank you for your valuable insights and guidance, which have greatly improved the quality of our research.

---

## [Decision Letter · Decision Letter 2]

29 Apr 2025

From physical self-esteem to sports participation: the mediating role of exercise motivation and social support in adolescents

PONE-D-24-57757R2

Dear Dr. wang,

We’re pleased to inform you that your manuscript has been judged scientifically suitable for publication and will be formally accepted for publication once it meets all outstanding technical requirements.

Kind regards,

Gal Harpaz, Ph.D.

Academic Editor

PLOS ONE

Additional Editor Comments (optional):

Dear Author

I am pleased to inform you that your manuscript, has been accepted for publication in PLOS ONE.

Your careful revisions and thoughtful responses to the reviewers' comments have strengthened the paper, and we look forward to sharing your important contribution with the broader scientific community.

Thank you for choosing PLOS ONE for your publication.

Please stay tuned for the next steps regarding production and publication scheduling.

Best regards,

Dr. Gal Harpaz

Academic Editor

PLOS ONE

Reviewers' comments:

Reviewer's Responses to Questions

**Comments to the Author**

1. If the authors have adequately addressed your comments raised in a previous round of review and you feel that this manuscript is now acceptable for publication, you may indicate that here to bypass the “Comments to the Author” section, enter your conflict of interest statement in the “Confidential to Editor” section, and submit your "Accept" recommendation.

Reviewer #3: All comments have been addressed

2. Is the manuscript technically sound, and do the data support the conclusions?

Reviewer #3: Yes

3. Has the statistical analysis been performed appropriately and rigorously? 

Reviewer #3: Yes

4. Have the authors made all data underlying the findings in their manuscript fully available?

Reviewer #3: Yes

5. Is the manuscript presented in an intelligible fashion and written in standard English?

Reviewer #3: Yes

6. Review Comments to the Author

Reviewer #3: (No Response)

7. PLOS authors have the option to publish the peer review history of their article (what does this mean? ). If published, this will include your full peer review and any attached files.

**Do you want your identity to be public for this peer review?** For information about this choice, including consent withdrawal, please see our Privacy Policy .

Reviewer #3: No

---

## [Editor Report · Acceptance letter]

PONE-D-24-57757R2

PLOS ONE

Dear Dr. wang,

I'm pleased to inform you that your manuscript has been deemed suitable for publication in PLOS ONE. Congratulations! Your manuscript is now being handed over to our production team.

Kind regards,

on behalf of

Dr. Gal Harpaz

Academic Editor

PLOS ONE